# Incorporation of Hybrid Nanomaterial in Dental Porcelains: Antimicrobial, Chemical, and Mechanical Properties

**DOI:** 10.3390/antibiotics10020098

**Published:** 2021-01-20

**Authors:** Carla L. Vidal, Izabela Ferreira, Paulo S. Ferreira, Mariana L. C. Valente, Ana B. V. Teixeira, Andréa C. Reis

**Affiliations:** Dental Materials and Prosthesis Department, Ribeirão Preto School of Dentistry, University of São Paulo, Av. do Café, s/n, Ribeirão Preto 14040-904, Brazil; carla.vidal@usp.br (C.L.V.); izabela.ferreira@usp.br (I.F.); b14vilela@hotmail.com (P.S.F.); mariana.lima.valente@usp.br (M.L.C.V.); ana.beatriz.teixeira@usp.br (A.B.V.T.)

**Keywords:** antimicrobials, biofilms, biomaterials, dental porcelain, nanoparticles

## Abstract

Biofilm formation on biomaterials is a challenge in the health area. Antimicrobial substances based on nanomaterials have been proposed to solve this problem. The aim was to incorporate nanostructured silver vanadate decorated with silver nanoparticles (*β*-AgVO_3_) into dental porcelains (IPS Inline and Ex-3 Noritake), at concentrations of 2.5% and 5%, and evaluate the surface characteristics (by SEM/EDS), antimicrobial activity (against *Streptococcus mutans*, *Streptococcus sobrinus*, *Aggregatibacter actinomycetemcomitans*, and *Pseudomonas aeruginosa*), silver (Ag^+^) and vanadium (V^4+^/V^5+^) ions release, and mechanical properties (microhardness, roughness, and fracture toughness). The *β*-AgVO_3_ incorporation did not alter the porcelain’s components, reduced the *S. mutans, S. sobrinus* and *A. actinomycetemcomitans* viability, increased the fracture toughness of IPS Inline, the roughness for all groups, and did not affect the microhardness of the 5% group. Among all groups, IPS Inline 5% released more Ag^+^, and Ex-3 Noritake 2.5% released more V^4+^/V^5+^. It was concluded that the incorporation of *β*-AgVO_3_ into dental porcelains promoted antimicrobial activity against *S. mutans*, *S. sobrinus*, and *A. actinomycetemcomitans* (preventing biofilm formation), caused a higher release of vanadium than silver ions, and an adequate mechanical behavior was observed. However, the incorporation of *β*-AgVO_3_ did not reduce *P. aeruginosa* viability and increased the surface roughness of dental porcelains.

## 1. Introduction

Advances in the development of biomaterials with antimicrobial effects reduce biofilm formation and prevent disease. Antibiotic substances can promote bacterial resistance, thus, the nanoparticles are an alternative with greater chemical reactivity and a smaller size to penetrate biofilm [1]. Vanadates, compounds that contain vanadium in its highest oxidation state, allow intercalation with other elements, resulting in the synthesis of nanostructures with different compositions and properties. Recently, nanostructured silver vanadate decorated with silver nanoparticles (*β*-AgVO_3_), a hybrid system based on vanadate nanowires and silver nanoparticles, was introduced [2]. This nanomaterial dissociates into silver (Ag^+^) and vanadium (V^4+^ and V^5+^) ions and acts on bacterial structures [2,3].

In dentistry, *β*-AgVO_3_ was incorporated into several dental materials, such as acrylic resins, denture liners, impression materials, endodontic sealers, and prosthetic components, and demonstrated antimicrobial activity against *Streptococcus mutans*, *Candida albicans*, *Staphylococcus aureus*, *Pseudomonas aeruginosa*, *Enterococcus faecalis*, and *Escherichia coli* [4,5,6,7,8,9,10,11]. A recent study that incorporated *β*-AgVO_3_ into dental porcelains indicated a promising antibacterial effect against *S. mutans* through the agar diffusion method, and the *β*-AgVO_3_ addition did not alter the Vickers microhardness in relation to the control group [12]. However, studies that prove the antimicrobial effectiveness against several species of microorganisms and the influence of this nanomaterial on the other characteristics and properties of these dental porcelains have not yet been realized.

Ceramic materials can be applied in dentistry to manufacture dental crowns over teeth or implants, fixed prosthesis, or osseointegrated implants. The incorporation of antimicrobial agents into these materials can prevent failures in prosthetic rehabilitation, such as secondary caries (initiated by microorganisms like *Streptococcus mutans* in 40% of cases [13,14,15,16]), periodontal disease [17], and even peri-implantitis [18]. Some attempts to add antimicrobials to dental ceramics have been made with the incorporation of silver nanoparticles in feldspathic porcelain [19], and into zirconia for dental implant applications [20], antimicrobial glass coatings applications [21], and aluminum particles in ceramic compounds [22].

This study proposes the incorporation of *β*-AgVO_3_ initially into dental porcelain, given the antibacterial potential observed, with prospects of application in other ceramic systems that expand the possibilities of prosthetic and implant rehabilitation. Thus, the purpose of the present study was to incorporate *β*-AgVO_3_ into dental porcelains and investigate the surface characteristics, and the antimicrobial, chemical, and mechanical properties. The hypothesis tested was that *β*-AgVO_3_ incorporation would influence the surface characteristics, and the antimicrobial, chemical, and mechanical properties of dental porcelains tested.

## 2. Results

### 2.1. Surface Characteristics and Chemical Composition

The control group of IPS Inline feldspathic porcelain showed few irregularities on the surface, with some aluminum and silicon peaks (Figure 1a). When *β*-AgVO_3_ was incorporated, greater surface roughness was observed. For IPS Inline 2.5%, deeper pores and peaks were observed (Figure 1b), and for IPS Inline 5%, steeper peaks were observed (Figure 1c). The Ex-3 Noritake control group showed more peaks and pores than the IPS Inline control group (Figure 1d). With the *β*-AgVO_3_ incorporation, a considerable increase in surface irregularities was observed. The Ex-3 Noritake 2.5% presented high surface roughness and surface morphology distinct from the control group (Figure 1e), and the 5% group showed more peaks and deeper pores (Figure 1f). The elemental analysis by EDS showed higher oxygen and slightly lower potassium and zinc levels for IPS Inline when compared to Ex-3 Noritake (Appendix A). Both groups incorporated with 5% *β*-AgVO_3_ showed a higher percentage of silver and vanadium ions than groups incorporated with 2.5% *β*-AgVO_3_ (Appendix A). In general, the atomic percentage of chemical elements present in dental porcelains did not change with the incorporation of the nanomaterial.

### 2.2. Antimicrobial Activity

The minimum inhibitory concentration of the pure *β*-AgVO_3_ necessary to inhibit the growth of microorganisms *S. mutans*, *S. sobrinus*, and *A. actinomycetemcomitans* was 250 µg/mL, and for *P. aeruginosa*, was 31.25 µg/mL (Table 1). When *β*-AgVO_3_ was incorporated into the IPS Inline, it caused a reduction in cell viability of *S. mutans*, at 5% concentrations (*p* < 0.05). Additionally, 5% *β*-AgVO_3_ also reduced the metabolic activity of *S. sobrinus* when incorporated into Ex-3 Noritake, and reduced the colony-forming units per milliliter (CFU/mL) when incorporated into the IPS Inline (*p* < 0.05). The incorporation of 2.5% *β*-AgVO_3_ into IPS Inline showed antibacterial effects on the metabolic activity of *A. actinomycetemcomitans* (*p* < 0.05). The other concentrations of *β*-AgVO_3_ incorporated into IPS Inline and Ex-3 Noritake showed no antimicrobial effects against *A. actinomycetemcomitans* or *P. aeruginosa* (*p* > 0.05) (Table 1).

The photon images of biofilms formed on the surface of dental porcelains incorporated with different concentrations of *β*-AgVO_3_ are in agreement with the results observed in the microbiology analysis. In Figure 2, it was observed that the IPS Inline 2.5% *β*-AgVO_3_ group (Figure 2b) presented a thicker *S. mutans* biofilm, with clusters of non-viable cells surrounded by viable cells, showing a different pattern from the control group (Figure 2a). Clusters also were observed in the *S. sobrinus* biofilm formed on the surface of the 5% IPS Inline group (Figure 2d), and with a higher proportion of non-viable cells compared to the control group (Figure 2c). On the surface of the IPS Inline 2.5% group (Figure 2f), a lower proportion of viable cells was observed than in the control group (Figure 2e) for a biofilm of *A. actinomycetemcomitans*. For the Ex-3 Noritake groups, a thicker *S. sobrinus* biofilm was observed with the incorporation of 5% *β*-AgVO_3_ (Figure 3b), with a higher proportion of non-viable cells, compared to the control group (Figure 3a). Despite the quantitative results for *P. aeruginosa*, the photon images demonstrated a higher proportion of non-viable cells in Ex-3 Noritake incorporated with 5% (Figure 3d), compared to the control group (Figure 3c).

### 2.3. Silver and Vanadium Ions Release

The group incorporated with 5% *β*-AgVO_3_ into IPS Inline presented the highest Ag^+^ release (*p* < 0.05). Ex-3 Noritake porcelain incorporated with 2.5% nanomaterial presented the highest release of V^4+^/V^5+^ (*p* < 0.05) in 7 days. No statistical difference was found for the release of V^4+^/V^5+^ between the groups modified by the nanomaterial at 30 and 120 days (*p* > 0.05). In general, the groups released more V^4+^/V^5+^ than Ag^+^, and within the same group, there were no variations in the release of V^4+^/V^5+^ or Ag^+^ over time (*p* > 0.05) (Table 2).

### 2.4. Mechanical Tests

The incorporation of *β*-AgVO_3_ into dental porcelains influenced the mechanical behavior of these materials. The surface roughness increased with the nanomaterial incorporation; the 5% *β*-AgVO_3_ group was statistically different from the IPS Inline (*p* = 0.045) and Ex-3 Noritake (*p* = 0.036) control groups. The incorporation of 2.5% *β*-AgVO_3_ decreased the microhardness compared to the control (*p* < 0.001) and 5% (*p* = 0.014) groups, which did not differ from each other. In the IPS Inline, the *β*-AgVO_3_ incorporation increased the fracture toughness, mainly for the 5% group (*p* < 0.001). For Ex-3 Noritake, there was no difference between the groups (*p* = 1.000) (Table 3).

## 3. Discussion

This study used two commercial feldspathic dental porcelains, IPS Inline and Ex-3 Noritake, to evaluate the viability of the mixture between ceramic materials and the *β*-AgVO_3_ nanomaterial in an unprecedented way. One of the biggest challenges in incorporating antimicrobial compounds into ceramic materials is that the material is exposed to high temperatures during processing, which may impair the action of the added compound [23]. Positive results were obtained in this study once the chemical composition confirmed the viability of the proposed mixture with the unaltered atomic proportion of the porcelain elements, adding to the beneficial antimicrobial effects of the *β*-AgVO_3_ addition, without major damage to the mechanical properties. Thus, the null hypothesis tested in this study was rejected.

The *β*-AgVO_3_ antimicrobial effects observed were due to the action of silver (Ag^+^) and vanadium (V^4+^ and V^5+^) ions released by the nanomaterial in contact with bacterial surfaces [2,3]. Ag^+^ adheres to the cytoplasmic membrane or cell wall through electrostatic interaction with sulfur proteins, which can permeate the membrane and cause its rupture [24]. Vanadium ions in a pentavalent state of oxidation (V^5+^) and Ag^+^ interact with the thiol groups present in the enzyme’s bacterial metabolism, forming stable complexes. Ag^+^ still comes into contact with bacterial DNA, preventing their replication, and oxidation-reduction between V^4+^ and V^5+^ leads to oxidative stress in the bacterial cell [3].

The pure *β*-AgVO_3_ demonstrated antimicrobial activity, confirming its potential to be used as an antimicrobial agent, as observed in other studies [4,9]. When incorporated into dental porcelains, it presented more antibacterial effects against *S. mutans* and *S. sobrinus*, which can prevent the incidence of dental caries. Oda et al. [25] related that in the presence of *S. mutans* and *S. sobrinus*, the incidence of dental caries is significantly higher than when in the presence of *S. mutans* only. The IPS Inline with 5% *β*-AgVO_3_ presented higher action against *S. mutans* than *S. sobrinus*, which can be due to the greater resistance of the *S. sobrinus*, whose virulence and acid-forming capacity is higher and faster than that of *S. mutans* [26,27]. A positive effect against *S. sobrinus* was observed in Ex-3 Noritake incorporated with 5% *β*-AgVO_3_.

IPS Inline with 2.5% *β*-AgVO_3_ showed antimicrobial activity against *A. actinomycetemcomitans*. This microorganism is commonly found in aggressive periodontal disease [28,29], and may cause extraoral infections, such as infectious endocarditis, bacterial arthritis, pregnancy-associated septicemia, brain abscesses, and osteomyelitis, but its primary source is the oral cavity [29]. *Acinetobacter* spp. associated with *P. aeruginosa* participate in the etiopathogenesis of periodontal diseases, and *P. aeruginosa, A. actinomycetemcomitans*, *Acinebactor baumannii*, and species of the red complex (*Porphyromonas gingivalis*, *Tannerella forsythia*, and *Treponema denticola*) increase the likelihood of developing aggressive periodontitis [30]. Against *P. aeruginosa*, the porcelains incorporated with *β*-AgVO_3_ showed no antimicrobial activity. The resistance of *A. actinomycetemcomitans* and *P. aeruginosa*, both of which are Gram-negative, to the action of *β*-AgVO_3_ can be due to structural differences, as they have an additional external membrane that Gram-positive bacteria do not possess, resulting in different susceptibility and permeability to antimicrobial agents [31].

The *β*-AgVO_3_ incorporated into dental porcelains showed lower antimicrobial activity than the pure *β*-AgVO_3_ for *P. aeruginosa*. This can be due to the total non-release of the *β*-AgVO_3_’s components. The ions release results showed that all groups released more vanadium ions than silver ions. However, in the synthesis of *β*-AgVO_3_, a greater amount of silver was added (2 mmol of silver nitrate) compared to the amount of vanadium (1 mmol of ammonium metavanadate) [3], and thus, a higher release of silver ions was expected, which may have compromised the antimicrobial activity of the dental porcelains. This greater release of vanadium than silver was also observed in other dental materials incorporated with *β*-AgVO_3_ [32,33].

The results also demonstrated that the IPS Inline presented better antimicrobial effects than the Ex-3 Noritake. This difference can be due to the greater release of silver and vanadium ions by IPS Inline, and due to differences in this porcelain’s sintering cycle led to a change in the nanomaterial’s properties. During processing, Ex-3 Noritake reaches temperatures of 600 °C, as indicated by the manufacturer, whereas IPS Inline is limited to 403 °C. The high temperatures produced during porcelain sintering negatively impact the incorporation of antimicrobial agents [23]. In addition, the porcelain’s composition (Ex-3 Noritake presented less K and more Zn wt.% elements) and particle type may not have interacted or integrated with the nanomaterial, and may have therefore inhibited effective antimicrobial release.

The *β*-AgVO_3_ incorporation changed the surface characteristics and higher roughness was observed. Ferreira et al. [12] also observed an increase in roughness with the addition of *β*-AgVO_3_ into IPS Inline, mainly in the 10% group, which was not included in this study for this reason. This higher roughness may have provided biofilm retention in the groups that did not show a reduction in the bacterial contingent, such as the increased metabolic activity in *P. aeruginosa* for the IPS Inline and Ex-3 Noritake modified groups. The roughness has a considerable influence on biofilm formation because it favors a greater quantity and early maturation of bacteria that are lodged in the pores [34]. Despite the higher roughness, the nanomaterial released demonstrated antimicrobial effects in some groups.

Roughness is also related to mechanical microretention between the cementing agent and the porcelain, and is positively correlated with bond strength [35]. The *β*-AgVO_3_ incorporation also caused a change in porcelain color, especially after the firing. Thus, the proposed new material can be used on the inner face of the restorations, so as not to interfere in the final color, and it may also favor their mechanical retention due to the roughness [36].

The photomicrographs of surface characteristics suggest that the incorporation interfered more in the material conformation when compared to the control groups and indicate that this interference was different from one porcelain to another, which corroborates the mechanical and antimicrobial results. Morphology may also have interfered with antimicrobial potential due to nanomaterial dispersion. When used in conjunction with SEM, dispersive EDS is a chemical microanalysis technique that facilitates the investigation of various samples’ compositional details [37]. Based on this analysis, the porcelains presented few differences in oxygen, potassium, and zinc levels. The presence of silver and vanadium in the groups incorporated with *β*-AgVO_3_ indicates the presence of the nanomaterial in the new material’s structure, without changing the components inherent to the porcelain itself.

Several parameters, such as microhardness and fracture toughness, influence the porcelain’s mechanical properties and clinical use. A lower fracture toughness value can lead to poorer clinical performance [38]. When this property was evaluated, the incorporation of the nanomaterial in IPS Inline promoted an increase in fracture toughness, which can lead to a considerable increase in the material’s ability to absorb elastic energy, indicating a greater chance of surviving more severe impacts [39]. The fracture toughness values for the modified groups of IPS Inline are in accordance with results shown in the literature for feldspathic porcelains [39]. In the microhardness evaluation, the group incorporated with 2.5% *β*-AgVO_3_ presented the lowest value. However, this property did not change in the group incorporated with 5% *β*-AgVO_3_, demonstrating that the incorporation of higher concentrations of vanadate does not harm the mechanical properties, and still favors the antimicrobial effectiveness. The surface roughness and microhardness values for IPS Inline were published by Ferreira et al. [12]. This work compared the IPS Inline values with the Ex-3 Noritake values.

It was observed that Ag^+^ and V^4+^/V^5+^ ions release decreased in 120 days for some groups. It was not the purpose of this study to quantify the ions released by the dental porcelains, however, the Ag^+^ and V^4+^/V^5+^ are cations that may have attached to the elements released by the porcelains, as anions released by silicate (SiO_3_^2−^), for example. The Ag^+^ and V^4+^/V^5+^ ions release is important to the antimicrobial activity, however, and, depending on the quantity, can be harmful to health. The cytotoxicity of the pure *β*-AgVO_3_ was investigated against *Daphnia similis*, an aquatic organism, and its cytotoxic effect was attributed to the silver [33]. When incorporated into acrylic resin, Castro et al. [32] reported that *β*-AgVO_3_ presents a little cytotoxicity to mouse fibroblasts (L929) if used at low concentrations, due to the lower release of silver and vanadium ions. In this study, a low concentration of Ag^+^ was detected, favoring the biocompatibility; however, this property should be investigated.

Significant results were obtained by incorporating *β*-AgVO_3_ into dental porcelains, including imbuing materials with antimicrobial properties against microorganisms frequently associated with dental problems. Thus, the antimicrobial action identified in this study has high significance, as the nanomaterial was able to maintain its effectiveness after porcelain processing. Improving the fracture toughness property of IPS Inline and maintaining this property for Ex-3 Noritake through the incorporation of the nanomaterial suggests the possible viability of this material for clinical application. The porcelains used are commercial formulations, so this study is an initial analysis of the viability of mixing porcelain materials and *β*-AgVO_3_. Therefore, the development of specific formulations to receive this additive should be evaluated in the future, including the application of *β*-AgVO_3_ in other ceramic systems that expand the possibilities of prosthetic and implant rehabilitation.

This study has its limitations because it was performed strictly in vitro. The interaction between microorganisms in vivo may lead to different results, and further analysis is important to consider the clinical use of these porcelains. The adhesive strength of these specimens to the enamel should also be evaluated in future studies.

Although the results regarding the mechanical properties were altered, the literature is not clear about the maximum acceptable alteration levels. Because the suggested use of this material is limited to the internal regions of prostheses, achieving antimicrobial results after firing processing, as well as processing the modified material itself and giving it shape, contour, and strength, shows the great advancement and potential use that this methodological proposal offers.

## 4. Materials and Methods

### 4.1. Specimen Preparation

The synthesis of the nanostructured silver vanadate decorated with silver nanoparticles (*β*-AgVO_3_) was performed according to Holtz et al. [3] and Castro et al. [4]. Briefly, 0.9736 g of ammonium metavanadate (NH_4_VO_3_; 99%; Merck KGaA, Darmstadt, Germany) and 1.3569 g of silver nitrate (AgNO_3_; 99.8%; Merck KGaA) were solubilized separately in 200 mL of distilled water. One solution was added to the other, obtaining the silver vanadate solution, which was vacuum filtered, washed with distilled water and absolute ethanol, and dried in a vacuum for 10 h to obtain the *β*-AgVO_3_ powder. The nanoparticle’s characterization by X-ray diffraction (XRD) was verified by Castro et al. [6], and the *β*-AgVO_3_ presented the beta phase. The powder obtained after the synthesis was analyzed by scanning transmission electron microscopy (Magellan 400L; FEI Company, Hillsboro, OR, USA) to confirm the presence of the silver nanoparticles on the surface of the nanowires (Figure 4).

Two commercial dental porcelains were used: IPS Inline (Ivoclar Vivadent, Schaan, Liechtenstein) and Ex-3 Noritake (Noritake Kizai CO., Nagoya, Japan) in A3 color for dentin. The *β*-AgVO_3_ was added in concentrations of 2.5 and 5 wt%, based on previous studies [4,5,6,7,8,9,10,11,12]. The entire contents of the dental porcelain bottle were considered 100%, and the masses (wt%) corresponding to the nanomaterial concentrations (2.5% and 5%) were subtracted from the porcelain powder, and the *β*-AgVO_3_ powder was added. A control group was obtained without *β*-AgVO_3_. The same amount of liquid was used for all groups. The powder was mixed with the liquid and the mix was deposited in a Teflon matrix with a circular shape (8 mm diameter × 2 mm thick), then removed and prepared for oven processing. The specimens were fired over refractory wool (initial temperature of 403 °C, heating rates of 60 °C/minute; firing temperature of 910 °C for IPS Inline and initial temperature of 600 °C, heating rates of 45 °C/minute; firing temperatures of 920 °C for Ex-3 Noritake). After firing, the furnace door was opened only 10% until the temperature inside the oven reached 300 °C, whereupon it was opened completely. The specimens were then submitted to a second firing (900 °C), following a fast cooling protocol (45 and 60 °C/second respectively). Thus, six groups were obtained: IPS Inline control (commercial porcelain, without *β*-AgVO_3_), IPS Inline with 2.5% *β*-AgVO_3_, IPS Inline with 5% *β*-AgVO_3_, Ex-3 Noritake control (commercial porcelain, without *β*-AgVO_3_), Ex-3 Noritake with 2.5% *β*-AgVO_3_, Ex-3 Noritake with 5% *β*-AgVO_3_. After, they were polished with water sandpaper for surface planning, and underwent gas sterilization with ethylene oxide (Acecil, Campinas, Brazil) (Table 4).

### 4.2. Surface Characteristics and Chemical Composition

The surface characterization of the specimens (*n* = 2) and elemental microanalysis was carried out using energy-dispersive X-ray spectroscopy (EDS—IXRF Systems mod. 500 Digital Processing, Houston, TX, USA) coupled to a Scanning Electron Microscopy (SEM—ZEISS model EVO 50, Cambridge, United Kingdom) with a 20 kV electron beam, using SE (secondary electrons) and BSD (backscattered electron) detectors for topographic and compositional evaluations, respectively. The specimens were gold-sputtered for 120 s on BAL-TEC equipment (model SCD 050 Sputter Coater, Furstentum, Liechtenstein). Microanalysis was performed at a working distance of 8.5 mm, Iprobe at 20 nA, and dead time at approximately 30% in 300× magnification, and the photomicrographs were obtained at 1000× magnifications.

### 4.3. Antimicrobial Activity

The minimum inhibitory concentration (MIC) of the pure *β*-AgVO_3_ powder against *Streptococcus mutans* (ATCC 25175), *Streptococcus sobrinus* (ATCC 33402), *Aggregatibacter actinomycetemcomitans* (ATCC 33384)*,* and *Pseudomonas aeruginosa* (ATCC 27853) was performed by the method of successive dilutions described by the Clinical and Laboratory Standards Institute in 96-well plates [40]. The plates were prepared with the culture medium supplemented with the microorganism. Ten decreasing concentrations of the nanomaterial were obtained, starting with a concentration of 0.5 mg/mL, and each dilution reduced the concentration by half. Two control groups were obtained: one positive, with microorganism and culture medium, and the other negative, with *β*-AgVO_3_ and sterile culture medium. After 24 h of incubation, the bacteriostatic effect of the material was assessed by the turbidity of the culture visible to the naked eye [4].

The antimicrobial activity of the dental porcelains incorporated with 2.5% and 5% *β*-AgVO_3_ was evaluated by colony-forming units per milliliter (CFU/mL) and metabolic activity. For biofilm formation, microorganisms were obtained from a recent culture and standardized in spectrophotometer (PCB 687, BYK Gardner GMBH, Geretsried, Germany) at a wavelength of 625 nm and a concentration of 10^8^ CFU/mL bacteria. Specific culture mediums for each microorganism were used (modified SB-20 for *S. mutans*, *S. sobrinus*, and *P. aeruginosa*, and Tryptic Soy Broth for *A. actinomycetemcomitans*). Specimens (*n* = 10) were inserted into 24-wells plates and 1000 μL of the culture medium inoculated was added to each well. A negative control group was obtained with one specimen from each group and a sterile culture medium, to check the sterility of the samples. The plates were incubated at 37 °C for 1 h 30 min under stirring at 750 rpm (Shaker, Cienlab, Campinas, SP, Brazil) for adhesion. Then, the specimens were rinsed with a saline solution and 1000 μL of sterile culture medium was added. The plates were incubated for 48 h under the same conditions for growth and maturation biofilm [5].

The cell viability of the microorganisms was quantified by colony-forming units per milliliter (CFU/mL) and metabolic activity. For CFU/mL, after biofilm formation, each specimen was washed, added to a microtube containing 1000 μL of Phosphate Buffered Saline, and sonicated in an ultrasonic bath (Altsonic, Clean 9CA, Ribeirão Preto, SP, Brazil) (200 watts/40 Hz) for 20 min. Afterward, 25 μL was collected and serial dilutions (10^−1^ to 10^−4^) were seeded in specific culture medium and incubated at 37 °C for 24 h (*n* = 10). The CFU/mL were counted, and the data were normalized in Log_10_. The metabolic activity was evaluated by an XTT Cell Viability Assay Kit (Uniscience, São Paulo, Brazil). After biofilm formation, each specimen was washed and 50 μL of XTT solution was placed in each well (*n* = 10). The plates were incubated in the dark at 37 °C for 2 h. Then, 100 μL aliquots of the homogenized suspension from each well were transferred to 96-well plates, in triplicate, and the absorbance was quantified by spectrophotometry at 492 nm, using a microplate reader (Synergy II, BioTek Instruments, Winooski, VT, USA) [5].

Qualitative analysis of the biofilms was performed using multiphoton microscopy. After biofilm formation, each specimen was washed (*n* = 2) and stained with 300 μL of Live/Dead^®^ BacLight ™ L 7007 dye (Molecular Probes, Inc., Eugene, OR, USA) for 15 min, protected from light. The specimens were evaluated using a Zeiss LSM-7MP multiphoton microscope (Carl Zeiss, Oberkochen, Germany). Two-photon images were acquired, selecting an area of 4 × 4 mm at the center of the biofilm, and then analyzed, using ZEN LSM (Carl Zeiss) and Image J (National Institutes of Health, Bethesda, MD, USA) software.

### 4.4. Silver and Vanadium Ions Release

The release of silver (Ag^+^) and vanadium (V^4+^/V^5+^) ions was assessed by atomic absorption spectroscopy (Varian AA240FS, Varian Inc., Palo Alto, California, USA). The specimens (*n* = 3) were immersed in tubes with 10 mL of deionized water, kept at 37 °C under agitation in an orbital shaker (Tecnal TE 420, Tecnal Scientific Equipment, Piracicaba, São Paulo, Brazil), with 6 h of agitation per day, for 7, 30 and 120 days. The Ag^+^ and V^4+^/V^5+^ ions were quantified by collecting 1 mL of each tube by calibration curves built in the equipment.

### 4.5. Mechanical Tests

For the evaluation of mechanical behavior, the surface roughness (*n* = 10) was carried out using the rugosimeter Surftest SJ-201P (Mitutoyo Corporation, Kawasaki, Japan), according to technical standard NBR ISO 4287:2002. The Ra parameter was analyzed, a tip touched the piece, and covered a distance of 4 mm, making three measurements on each specimen in the direction of its larger diameter. The surface microhardness (*n* = 10) was measured through a microdurometer (Shimadzu HMV-2, Kyoto, Japan) with a Vickers type indenter with a load of 0.2 Kgf for 20 s. Three readings were performed in different regions for each specimen. The Vickers indentation method was used for analyzed fracture toughness (*n* = 10). Each print featured two pairs of radial cracks. Thirty pairs of perfect cracks were used (without polishing imperfections and trajectory deviations). Fracture toughness values were calculated using the equation proposed by Antis: *K_IC_* = 0.016 (*E/H*)^1/2^
*P/C*^3/2^, where *K_IC_* = material fracture toughness (MPa·m^1/2^); *E* = modulus of elasticity of material (GPa); *H* = material hardness (GPa); l = crack length (m); *P* = applied load (N); *C* = 1 + a (m); *a* = semi-diagonal of the Vickers impression (m).

### 4.6. Statistical Analysis

After verifying the distribution of data using the Shapiro–Wilk test (α = 0.05), the Kruskal–Wallis and Dunn’s post-hoc statistical tests were applied for antimicrobial analysis and ions release, and ANOVA and Bonferroni’s post-hoc for mechanical analysis. The SEM/EDS and multiphoton microscopy data were analyzed qualitatively.

## 5. Conclusions

*β*-AgVO_3_ incorporation into dental porcelains did not alter the porcelain’s components. It promoted antimicrobial activity in IPS Inline against *S. mutans*, *S. sobrinus*, and *A. actinomycetemcomitans*, preventing biofilm formation on the internal regions of prostheses. Furthermore, it increased fracture toughness of IPS Inline, and did not affect microhardness of the 5% group, conferring an adequate mechanical behavior. The dental porcelains incorporated with *β*-AgVO_3_ released more vanadium than silver ions. However, the incorporation of *β*-AgVO_3_ increased the surface roughness of dental porcelains.

## Figures and Tables

**Figure 1 antibiotics-10-00098-f001:**
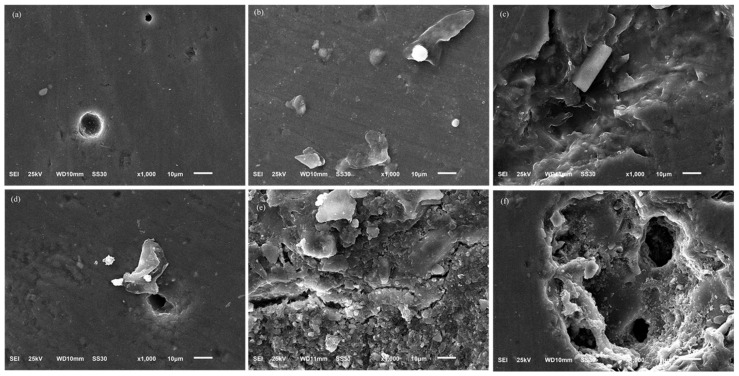
Photomicrographs of commercial dental porcelains incorporated with different concentrations of *β*-AgVO_3_ (Magnification: 1000×). (**a**) IPS Inline control. (**b**) IPS Inline incorporated with 2.5% *β*-AgVO_3_. (**c**) IPS Inline incorporated with 5% of *β*-AgVO_3_. (**d**) Ex-3 Noritake control. (**e**) Ex-3 Noritake incorporated with 2.5% *β*-AgVO_3_. (**f**) Ex-3 Noritake incorporated with 5% of *β*-AgVO_3_.

**Figure 2 antibiotics-10-00098-f002:**
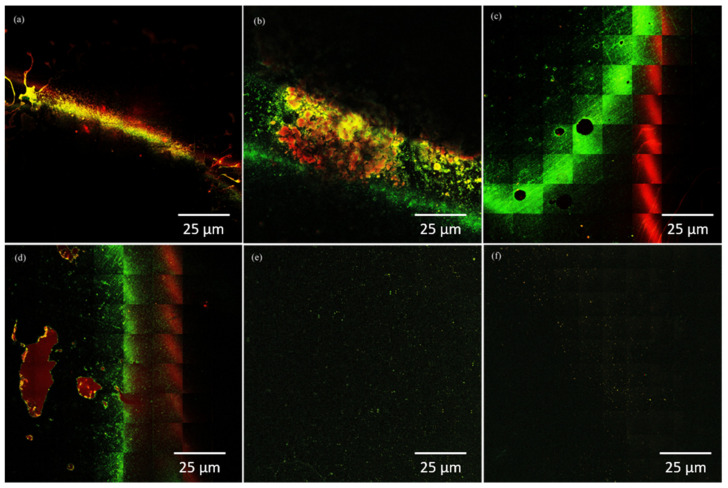
Photon images of biofilms of different microorganisms formed on the surface of dental porcelain IPS Inline incorporated with *β*-AgVO_3_, acquired by a multiphoton microscope. Green: viable cells (live); red: non-viable cells (dead). (**a**) *Streptococcus mutans* biofilm on the surface of the control group; (**b**) *S. mutans* biofilm on the surface of the 2.5% group; (**c**) *Streptococcus sobrinus* biofilm on the surface of the control group; (**d**) *S. sobrinus* biofilm on the surface of the 5% group; (**e**) *Aggregatibacter actinomycetemcomitans* biofilm on the surface of the control group; (**f**) *A. actinomycetemcomitans* biofilm on the surface of the 2.5% group.

**Figure 3 antibiotics-10-00098-f003:**
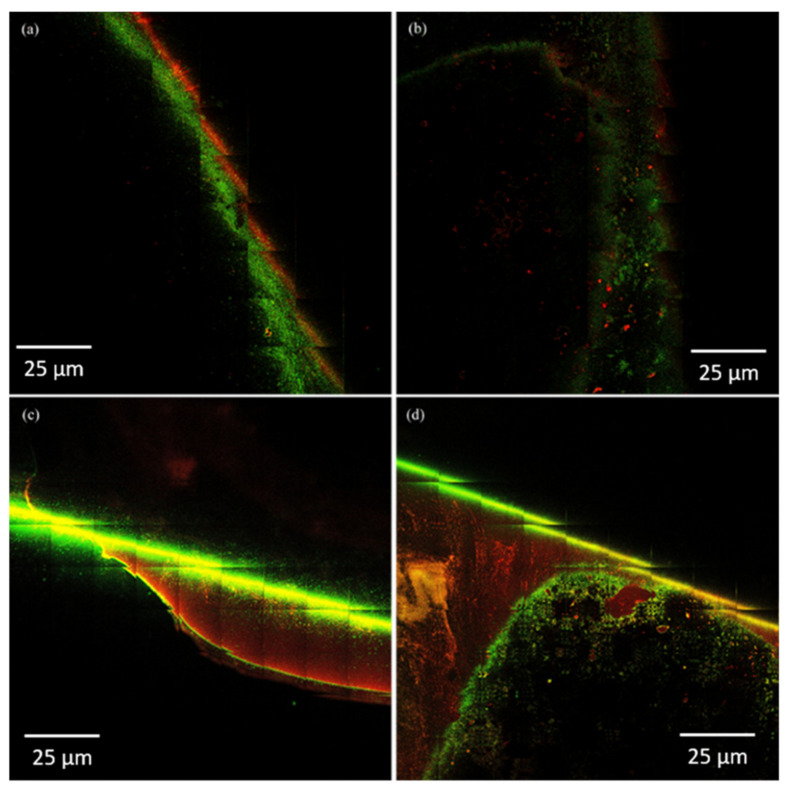
Photon images of biofilms of different microorganisms formed on the surface of dental porcelain Ex-3 Noritake incorporated with *β*-AgVO_3,_ acquired by multiphoton microscope. Green: viable cells (live); red: non-viable cells (dead). (**a**) *Streptococcus sobrinus* biofilm on the surface of the control group; (**b**) *S. sobrinus* biofilm on the surface of the 5% group; (**c**) *Pseudomonas aeruginosa* biofilm on the surface of the control group; (**d**) *P. aeruginosa* biofilm on the surface of the 5% group.

**Figure 4 antibiotics-10-00098-f004:**
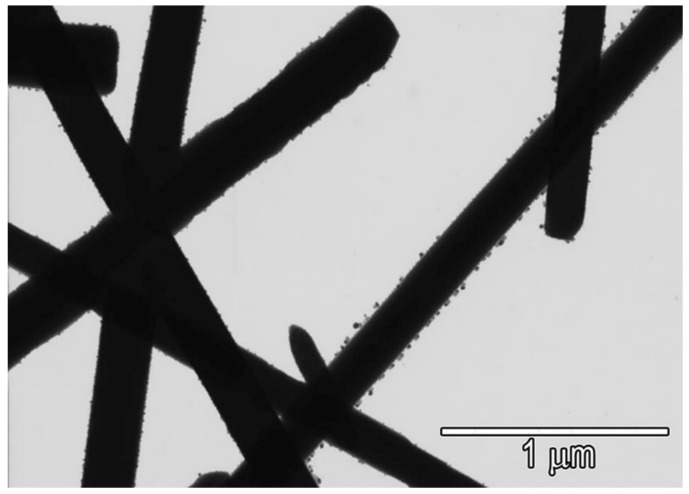
Scanning transmission electron microscopy micrograph of *β*-AgVO_3_ (×200,000 magnification). The silver vanadate nanowires presented a diameter of approximately 150 nm, coated with semi-spherical silver nanoparticles with dimensions of 25 nm.

**Table 1 antibiotics-10-00098-t001:** Minimum inhibitory concentration (MIC) of pure *β*-AgVO_3_ and antimicrobial activity of dental porcelains incorporated with *β*-AgVO_3_ on metabolic activity and colony-forming units per milliliter (CFU/mL) of microorganisms.

Microorganisms	MIC of *β*-AgVO_3_ (µg/mL)	Group	Metabolic Activity (Absorbance)		CFU/mL (Log_10_)	
			IPS Inline	Ex-3 Noritake	IPS Inline	Ex-3 Noritake
	250	Control	0.91 [0.77; 1.16] ^Aa^	0.24 [0.19; 0.49] ^Ba^	7.60 [7.25; 7.88] ^Aa^	6.78 [6.58; 7.23] ^Ba^
*Streptococcus mutans*		2.5%	0.67 [0.47; 0.95] ^a^	0.55 [0.45; 0.79] ^b^	5.71 [3.50; 7.16] ^ab^	7.40 [6.18; 7.95] ^a^
		5%	0.12 [0.06; 0.22] ^b^	0.22 [0.12; 0.30] ^a^	5.13 [4.03; 5.98] ^Ab^	7.04 [6.31; 7.24] ^Ba^
	250	Control	1.04 [0.82; 1.21] ^a^	0.97 [0.86; 1.08] ^a^	8.21 [7.74; 8.44] ^a^	8.14 [7.63; 8.69]
*Streptococcus* *sobrinus*		2.5%	1.17 [0.77; 1.40] ^Aa^	0.97 [0.70; 1.16] ^Ba^	8.19 [5.22; 9.02] ^ab^	7.82 [7.61; 8.28]
		5%	1.06 [0.76; 1.29] ^Aa^	0.82 [0.57; 1.01] ^Bb^	7.03 [4.45; 7.90] ^Ab^	7.89 [6.81; 8.47] ^B^
	250	Control	1.10 [1.06; 1.12] ^Aa^	1.17 [1.14; 1.24] ^Ba^	7.30 [7.03; 7.35] ^a^	7.17 [7.10; 7.27] ^ab^
*Aggregatibacter actinomycetemcomitans*		2.5%	1.00 [0.98; 1.05] ^Ab^	1.20 [1.16; 1.30] ^Ba^	7.37 [7.25; 7.48] ^ab^	7.44 [7.17; 7.55] ^a^
		5%	1.12 [1.03; 1.23] ^ab^	1.18 [1.16; 1.20] ^a^	7.39 [7.27; 7.47] ^Ab^	7.14 [7.03; 7.23] ^Bb^
	31.25	Control	0.23 [0.13; 0.41] ^a^	0.09 [0.06; 0.18] ^a^	6.82 [6.29; 7.61]	6.37 [6.11; 7.59]
*Pseudomonas* *aeruginosa*		2.5%	0.50 [0.35; 0.94] ^b^	0.92 [0.53; 1.14] ^b^	7.44 [6.63; 8.14]	7.19 [6.70; 8.07]
		5%	0.31 [0.28; 0.41] ^Aab^	0.58 [0.39; 0.75] ^Bc^	6.46 [6.11; 8.31]	6.17 [5.27; 7.59]

Kruskal–Wallis, Dunn’s post-hoc, and Mann–Whitney tests (α = 0.05). Median [confidence interval]. ^a,b,c^ Equal lowercase letters in the same column indicate statistical similarity (*p* > 0.05). ^A,B^ Equal uppercase letters in the same row for each method indicate statistical similarity (*p* > 0.05).

**Table 2 antibiotics-10-00098-t002:** Different times of silver and vanadium ions release (ppm) of dental porcelains incorporated with *β*-AgVO_3_.

		Ag^+^			V^4+^/V^5+^		
Porcelain	Group	7 Days	30 Days	120 Days	7 Days	30 Days	120 Days
Ex-3 Noritake	Control	0 ^Aa^	0 ^Aa^	0 ^Aa^	0 ^Aa^	0 ^Aa^	0 ^Aa^
	2.5%	0.22 [0.07; 0.43] ^Abc^	0.23 [0.10; 0.39] ^Abc^	0.25 [−0.14; 0.75] ^Abc^	14.65 [−1.44; 25.68] ^Ab^	10.92 [6.67; 14.49] ^Ab^	2.64 [0.63; 4.25] ^Ab^
	5%	0.26 [−0.05; 0.54] ^Abc^	0.12 [−0.04; 0.33] ^Abc^	0.16 [−0.13; 0.53] ^Abc^	2.52 [1.18; 4.47] ^Ac^	1.98 [−0.27; 4.82] ^Ab^	3.99 [0.58; 7.67] ^Ab^
IPS Inline	Control	0 ^Aab^	0 ^Aab^	0 ^Aab^	0 ^Aa^	0 ^Aa^	0 ^Aa^
	2.5%	0.32 [−0.56; 0.60] ^Abc^	0.08 [−0.40; 0.83] ^Abc^	0.08 [−0.21; 0.46] ^Abc^	4.79 [0.61; 7.53] ^Abc^	4.24 [−4.70; 16.20] ^Ab^	5.33 [−2.87; 13.11] ^Ab^
	5%	0.45 [−0.06; 1.20] ^Ac^	0.41 [0.02; 0.81] ^Ac^	0.53 [−0.25; 1.24] ^Ac^	5.95 [−6.06; 22.27] ^Abc^	12.71 [2.11; 20.13] ^Ab^	1.31 [−2.41; 6.93] ^Ab^

Kruska–Wallis and Dunn’s post-hoc tests (α = 0.05). Median [confidence interval]. ^a,b,c^ Equal lowercase letters in the same column indicate statistical similarity (*p* > 0.05). ^A^ Equal uppercase letters in the same row indicate statistical similarity for each ion (*p* > 0.05).

**Table 3 antibiotics-10-00098-t003:** Mechanical behavior of dental porcelains incorporated with *β*-AgVO_3_.

Porcelain	Group	Roughness (µm)	Microhardness (Kgf/cm^2^)	Fracture Toughness (MPa·m^1/2^)
Ex-3 Noritake	Control	1.74 (1.44) ^a^	609.50 (62.07) ^a^	0.18 (0.41) ^a^
	2.5%	3.04 (1.30) ^ab^	497.20 (65.55) ^b^	0.20 (0.41) ^a^
	5%	3.56 (2.01) ^b^	571.50 (86.04) ^a^	0.19 (0.65) ^a^
IPS Inline	Control	1.76 (0.72) ^a^	542.46 (23.96) ^a^	0.19 (0.03) ^a^
	2.5%	3.37 (1.35) ^ab^	503.10 (48.60) ^b^	1.86 (0.25) ^b^
	5%	3.51 (2.09) ^b^	535.83 (36.93) ^a^	2.40 (0.49) ^c^

ANOVA and Bonferroni’s post-hoc tests (α = 0.05). Mean and standard deviation. ^a,b,c^ Equal lowercase letters in the same column indicate statistical similarity for each dental porcelain (*p* > 0.05). The surface roughness and microhardness values for IPS Inline were published by Ferreira et al. [12].

**Table 4 antibiotics-10-00098-t004:** Specimen specifications and number of samples (*n*) by experiment.

Specimens Size	Groups	SEM/EDS	CFU/mL and Metabolic Activity	Qualitative Analysis of Biofilm	Ions Release	Roughness Microhardness Fracture Toughness
8 mm diameter × 2 mm thick	IPS Inline control	*n* = 2	*n* = 10	*n* = 2	*n* = 3	*n* = 10
	IPS Inline with 2.5% *β*-AgVO_3_					
	IPS Inline with 5% *β*-AgVO_3_ Ex-3 Noritake control Ex-3 Noritake with 2.5% *β*-AgVO_3_ Ex-3 Noritake with 5% *β*-AgVO_3_					

## Data Availability

The data presented in this study are available on request from the corresponding author.

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
