# Peer review of "Incorporation of Hybrid Nanomaterial in Dental Porcelains: Antimicrobial, Chemical, and Mechanical Properties"

_antibiotics, 2021, doi:10.3390/antibiotics10020098_

Round 1

Reviewer 1 Report

Dear authors, thank you for this piece of research which I had the opportunity to review.

This is a very well conducted study with a sound methodological design. I have only found some points to improve:

  1. Introduction, p.2 line 52: about periimplantitis, please provide reference
  2. Introduction: Please add some background about mechanical properties since the information you give about antibacterial properties is broad but there is nothing about mechanical ones.
  3. Materials and Methods: I would add a table or a diagram with detailed sample size (n per group for the different experiments).
  4. Did you check if there is a shade change after adding nanoparticles to the material? I think this is a special point to be considered
  5. Do you think there are any implications of the silver and vanadium release clinically? Does the literature say something about this? Please comment in discussion
  6. Would this addition of nanoparticles imply any kind of staining of the underlying or adyacent teeth?

Author Response

Reply to the Reviewers’ comments

Dear Editor and Reviewers,

We sincerely appreciate the comments made in our paper, which are very useful to improve our work. We have considered all comments on the revised version of the manuscript. In the following answers, the reviewers’ comments are in bold and our answers are in black. For the revised manuscript the changes in the text were highlight using the "Track Changes" function in Microsoft Word.

Assistant Editor

- We noticed that Figure 4 is not cited in the main text, so we would like to ask you to please cite the Figure 4 in the main text or delete it.

            We apologize, figure 4 was erroneously cited in the text as figure 1, on page 9, in the materials and methods section, line 294. We changed “figure 1” for “figure 4”.

Reviewer 1

- Introduction, p.2 line 52: about periimplantitis, please provide reference.

            We included the reference [18] about the biofilm which can cause peri-implantitis, on page 2, line 52. This reference was also included on the References list, on page 12, line 452: 18. Melo F, Nascimento C, Souza DO, Albuquerque Jr. RF. Identification of oral bacteria on titanium implant surfaces by 16S rDNA sequencing. Clin Oral Implants Res. 2017;28:697-703. https://doi.org/10.1111/clr.12865.

- Introduction: Please add some background about mechanical properties since the information you give about antibacterial properties is broad but there is nothing about mechanical ones.

            A background about mechanical properties of the IPS Inline porcelain incorporated with β-AgVO3, published by Ferreira et al. (reference 12), was included in the text, on page 2, line 44 and 45, which says: “A recent study that incorporated the β-AgVO3 into dental porcelains indicated a promising antibacterial effect against S. mutans through agar diffusion method, and the β-AgVO3 addition did not alter the Vickers microhardness in relation to the control group [12].” And in the discussion section, on page 8, line 221 to 223, which says: “Ferreira et al. [12] also observed an increase in roughness with the addition of the β-AgVO3 into IPS Inline, mainly of the 10% group, that for this reason was not included in this study.”

- Materials and Methods: I would add a table or a diagram with detailed sample size (n per group for the different experiments).

            A table (Table 4) was added as requested, containing the specimen’s specifications (size and groups), and the number of samples (n) by experiment. The table can be verified on page 10, in the materials and methods section, line 321 to 324.

- Did you check if there is a shade change after adding nanoparticles to the material? I think this is a special point to be considered.

            Yes, due to the mustard coloring of the nanomaterial, the porcelain presented color change, especially after the firing. We added this information on the discussion section, and suggest that the material will be used on the inner face of the restauration to not interfere on the final color. This information can be check on page 8, line 230 to 234, which says: “The β-AgVO3 incorporation also caused a change in porcelain color, especially after the firing. Thus, the proposed new material can be used on the inner face of the restorations, so as not to interfere in the final color and it may also favor their mechanical retention due to the roughness [36].”

- Do you think there are any implications of the silver and vanadium release clinically? Does the literature say something about this? Please comment in discussion.

            Yes, the silver and vanadium release can be impact in the human health, since these ions can be cytotoxic to cells. We added in the discussion section the information about two studies that addressed the cytotoxicity of the pure β-AgVO3 and incorporated into other material. This information can be check on page 9, line 264 to 272, which says: “The Ag+ and V4+/V5+ ions release it is important to the antimicrobial activity, however, depending of the quantity, can be harmful to health. The cytotoxicity of the pure β-AgVO3 were investigated against Daphnia similis, an aquatic organism, and your cytotoxic effect was attributed to the silver [33]. When incorporated into acrylic resin, Castro et al. [32] reported that the β-AgVO3 presents a little cytotoxicity to mouse fibroblasts (L929) if used at low concentrations, due to the lower release of silver and vanadium ions. In this study, a low concentration of Ag+ was detected, favoring the biocompatibility, however, this property should be investigated.”

- Would this addition of nanoparticles imply any kind of staining of the underlying or adjacent teeth?

            We did not assess the color change of underlying or adjacent teeth by the material proposed in the study, however, in another study conducted by our group  it was observed that the β-AgVO3 potentiated the color change in endodontically treated teeth when incorporated into endodontic sealers (Vilela Teixeira AB, Vidal C L, de Castro DT, da Costa Valente ML, Oliveira-Santos C, Alves OL, dos Reis AC. Effect of incorporation of a new antimicrobial nanomaterial on the physicalchemical properties of endodontic sealers. Journal of Conservative Dentistry 2017; 20:392-397). This is an important point that must be evaluated in future research.

Reviewer 2 Report

This is a very significant study in a very important matter: biofilm formation on biomaterials.

The aim of this study was to incorporate the nanostructured silver vanadate decorated with silver nanoparticles (β-AgVO3) into dental  porcelains (IPS Inline and Ex-3 Noritake), at different concentrations followed by evaluation of surface characteristics (by SEM/EDS), antimicrobial activity, silver (Ag+) and vanadium (V4+/V5+) ions release, and mechanical properties (microhardness, roughness, and fracture 16
toughness).

The authors concluded that the incorporation of β-AgVO3 at dental porcelains promoted antimicrobial activity against S. mutans, S. sobrinus, and A. actinomycetemcomitans, preventing the biofilm formation, caused higher release of vanadium than silver ions, and an adequate mechanical behavior . However, the incorporation of β-AgVO3 did not reduced P. aeruginosa viability and increased the surface roughness of dental porcelains.

I suggest some minor changes:

Line 50: S. mutans must be in italic.

Line 100 Pag 4 Table 1: Please correct all bacterial names. Put them in the correct form.

Line 177: Please explain the sentence "When incorporated into dental porcelains, it presented more effect against S. mutans and S. sobrinus, demonstrating a spectrum of action against Gram-positive and Gram-negative bacteria, which can prevent the incidence of dental caries." This is not clear for me with the results.

Please make sure the bacteria names are in a correct form throughout the text.

This is a very elegant study that, with minor changes, deserve to be published.

Author Response

Reply to the Reviewers’ comments

Dear Editor and Reviewers,

We sincerely appreciate the comments made in our paper, which are very useful to improve our work. We have considered all comments on the revised version of the manuscript. In the following answers, the reviewers’ comments are in bold and our answers are in black. For the revised manuscript the changes in the text were highlight using the "Track Changes" function in Microsoft Word.

Assistant Editor

- We noticed that Figure 4 is not cited in the main text, so we would like to ask you to please cite the Figure 4 in the main text or delete it.

            We apologize, figure 4 was erroneously cited in the text as figure 1, on page 9, in the materials and methods section, line 294. We changed “figure 1” for “figure 4”.

Reviewer 2

- Line 50: S. mutans must be in italic.

            We apologize for the error, we corrected and putted S. mutans in italics (on line 44).

- Line 100 Pag 4 Table 1: Please correct all bacterial names. Put them in the correct form.

            Thank you for the suggestion, we corrected and putted the microorganisms complete names in the Table 1, on page 4, line 100.

- Line 177: Please explain the sentence "When incorporated into dental porcelains, it presented more effect against S. mutans and S. sobrinus, demonstrating a spectrum of action against Gram-positive and Gram-negative bacteria, which can prevent the incidence of dental caries." This is not clear for me with the results.

            For better understanding and being faithful to the results, we rewrite this phrase. The change can be check on page 7, line 178, which says: “When incorporated into dental porcelains, presented more antibacterial effect against S. mutans and S. sobrinus, which can prevent the incidence of dental caries.”

- Please make sure the bacteria names are in a correct form throughout the text.

            Thank you for this suggestion, we checked the bacteria names throughout the text.